# Antioxidant Lipid Supplement on Cardiovascular Risk Factors: A Systematic Review and Meta-Analysis

**DOI:** 10.3390/nu16142213

**Published:** 2024-07-10

**Authors:** Sitong Wan, Wenbin Wu, Yan Zhang, Jian He, Xiaoping Wang, Peng An, Junjie Luo, Yinhua Zhu, Yongting Luo

**Affiliations:** 1College of Food Science and Engineering, Gansu Agricultural University, Lanzhou 730070, China; adeline7wan@163.com (S.W.); zhangyan@gsau.edu.cn (Y.Z.); 2Key Laboratory of Precision Nutrition and Food Quality, Department of Nutrition and Health, China Agricultural University, Beijing 100193, China; wwb091828@163.com (W.W.); an-peng@cau.edu.cn (P.A.); 3National Center of Technology Innovation for Dairy, Hohhot 010110, China; hejian@yili.com; 4Zhejiang Medicine Co., Ltd., Shaoxing 312366, China; wangxiaoping@zmc.top; 5Food Laboratory of Zhongyuan, Luohe 462300, China; 6State Key Laboratory for Quality Ensurance and Sustainable Use of Dao-di Herbs, Artemisinin Research Center, and Institute of Chinese Materia Medica, China Academy of Chinese Medical Sciences, Beijing 100700, China

**Keywords:** antioxidant functional lipids, cardiovascular disease, meta-analysis

## Abstract

The efficacy of functional lipids with antioxidant properties in reducing cardiovascular risk has not been consistent. Randomized controlled trials (RCTs) reporting estimates for the effects of antioxidant functional lipid supplementations on cardiometabolic risk factors were searched up to 1 May 2024. Overall, antioxidant lipid supplementations, compared with placebo, had favorable effects on systolic blood pressure (lycopene: −1.95 [−3.54, −0.36] mmHg), low-density lipoprotein cholesterol (n6 fatty acid: −0.39 [−0.71, −0.06] mmol/L; astaxanthin: −0.11 [−0.21, −0.01] mmol/L), high-density lipoprotein cholesterol (n3 fatty acid: 0.20 [0.13, 0.27] mmol/L; n6 fatty acid: 0.08 [0.01, 0.14] mmol/L; astaxanthin: 0.13 [0.05, 0.21] mmol/L), total cholesterol (n6 fatty acid: −0.24 [−0.37, −0.11] mmol/L; astaxanthin: −0.22 [−0.32, −0.12] mmol/L; beta-carotene: −0.13 [−0.23, −0.04] mmol/L), triglyceride (n3 fatty acid: −0.37 [−0.47, −0.28] mmol/L; astaxanthin: −0.46 [−0.83, −0.10] mmol/L), and fasting blood insulin (astaxanthin: −2.66 [−3.98, −1.34] pmol/L). The benefits of antioxidant lipid supplementations appeared to be most evident in blood pressure and blood lipids in participants with different cardiometabolic health statuses. Notably, n9 fatty acid increased triglyceride and hemoglobin A1C in the total population, which increases CVD risk. Antioxidant lipid supplementations ameliorate cardiometabolic risk factors, while their effect may depend on type and cardiometabolic health status. Long-term RCTs are needed to corroborate risk–benefit ratios across different antioxidant functional lipid supplementation settings.

## 1. Introduction

Cardiovascular diseases (CVDs) continue to be a major cause of morbidity and mortality worldwide despite advances in preventive and therapeutic medicine [1]. High levels of blood glucose, blood pressure, and lipids, including low-density lipoprotein cholesterol, total cholesterol, and triglyceride, have been identified over the past 50 years as modifiable risk factors that act as predictors of CVD events and are thus key targets for the primary prevention of CVDs [2,3,4,5].

Of the deaths or disabilities associated with CVDs and type 2 diabetes (T2D), a significant proportion is attributable to suboptimal dietary habits, with different types of functional lipids garnering substantial attention [6,7,8]. Functional lipids refer to a class of lipids with specific physiological functions, defined as fat-soluble substances required for the maintenance of human nutrition and health, as well as the prevention and treatment of certain consequential nutrient deficiencies and endogenous diseases, especially hypertension, T2D, and other CVDs [9]. Functional lipids mainly include polyunsaturated fatty acids such as linoleic acid, linolenic acid, eicosapentaenoic acid, and eicosahexaenoic acid, as well as phytosterols [9]. Polyunsaturated fatty acids are the mainstay of functional lipids research and are generally defined as fatty acids containing either two or more double carbons with a carbon length of 18 or more. Phytosterols are active components that are widely found in roots, stems, leaves, fruits, and seeds of plants and possess favorable antioxidant properties and anti-inflammatory effects [9].

Oxidative stress is a contributing factor to CVDs, which has long been recognized as being involved in the pathogenesis of CVDs [10]. The adverse effects of oxidative stress on the cardiovascular system are attributed to decreased nitric oxide availability, inflammatory responses, and lipid peroxidation through which reactive oxygen species disrupt the homeostasis of the vascular wall, leading to defective endothelium-dependent vasodilatation [11]. Once formed, ROS activates nuclear factor-kappa B, leading to transcriptional activation of more than 100 genes involved in the immune system and inflammatory response, such as tumor necrosis factor-α, interleukin-1β, and interleukin-6. Macrophages absorb oxidized low-density lipoprotein and then convert it to foam cells in vascular endothelial cells, leading to the development of atherosclerotic lesions [10,11]. Preclinical and epidemiological studies have suggested that dietary antioxidants such as vitamin E, carotenoids, and polyphenols inhibit oxidative stress, thereby providing cardiovascular protection. Physiologically, unsaturated fatty acids, phytosterols, and other functional lipids with antioxidant properties may contribute to cardiometabolic health by removing free radicals and decreasing inflammation and platelet activity while preserving endothelial cell homeostasis and cardiac function. Further, free radicals impair beta cell function and insulin sensitivity, resulting in hyperglycemia and insulin resistance, thereby predisposing to CVDs [12,13].

However, randomized clinical trials (RCTs) investigating the efficacy of antioxidant functional lipid supplementations in reducing cardiometabolic risk factors displayed inconsistent findings. Unsaturated fatty acid supplementations have been shown to have certain beneficial effects on blood glucose [14] and lipids [15,16,17] among patients with T2D in some but not all studies [18,19]. Moreover, lycopene [20,21], astaxanthin [22,23], and beta-carotene [24,25] showed inconsistent results in CVD risk factors. To reconcile the inconsistencies in the literature regarding the role of antioxidant functional lipid supplementations in the development of health outcomes, we conducted a systematic review and meta-analysis of all available RCTs to investigate the effect of interventions with functional lipids with antioxidant properties on CVD risk factors.

## 2. Methods

Per the Preferred Reporting Items for Systematic Reviews and Meta-Analyses guidelines [26], we registered the study protocol in the International Prospective Register of Systematic Reviews (PROSPERO CRD42024536111). The included RCTs in this meta-analysis received ethics approval from relevant institutional review boards.

### 2.1. Search Strategy

The PubMed, Web of Science, and Embase databases were searched for RCTs published up to the 1st of May 2024. To conduct the systematic search, the following terms were used: (“N3 fatty acid” OR “n-6 fatty acid” OR “n-9 fatty acid” OR “EPA” OR “Eicosapentaenoic Acid” OR “DHA” OR “Docosahexaenoic Acid” OR “linoleic acid” OR “gamma-linolenic acid” OR “oleic acid” OR “lycopene” OR “astaxanthin” OR “beta-carotene”) AND (“Blood glucose” OR “Blood sugar” OR “glycemic” OR “Blood lipids” OR “triglyceride” OR “cholesterol” OR “Blood pressure”).

### 2.2. Inclusion and Exclusion Criteria

RCTs assessing the effect of antioxidant functional lipid supplementations on blood pressure (systolic blood pressure [SBP]; diastolic blood pressure [DBP]), blood lipids (total cholesterol [TC]; high-density lipoprotein cholesterol [HDL-C]; low-density lipoprotein cholesterol [LDL-C]; and triglyceride [TG]), and glycemic parameters (fasting blood glucose [FBG]; hemoglobin [A1C]; and fasting blood insulin [FBI]) were included for further review.

Trials without randomization, relevant cardiometabolic outcomes, a corresponding placebo or control substance, or viable mean change and standard deviation (SD) were excluded. Additionally, studies with intervention durations of less than 1 week or included participants with severe cardiovascular diseases, mental disorders, or other severe diseases at baseline were excluded.

### 2.3. Study Selection

The literature search was conducted independently by two reviewers and relevant published articles were identified. All selected studies were re-examined by a third reviewer and discrepancies were resolved through group discussions.

### 2.4. Data Extraction

Information extracted from eligible studies included the last name of the first author, publication year, geographic location of the study population, study design, participant characteristics (number of participants, mean age, sex, and health status), intervention substance, control substance, intervention dose, and intervention duration. For studies that did not report the SD of the mean difference between the baseline and endpoint cardiometabolic risk factors, the SD of the mean difference was estimated by the formula SDchange = square root [(SDbaseline2 + SDendpoint2)/2] [27]. If the outcome data were presented as graphs, we used WebPlotDigitizer (https://automeris.io/WebPlotDigitizer/, accessed on 1 January 2024.) to estimate the values.

### 2.5. Quality Assessment

The evaluation of the risk of bias in the included trials was conducted per the recommendations of the Cochrane Collaboration Handbook [28], which included six domains: selection bias (random sequence generation and allocation concealment); performance bias (blinding of participants and personnel); detection bias (blinding of outcome assessment); attrition bias (incomplete outcome data); reporting bias (selective reporting), and other bias.

The GRADE (Grading of Recommendations, Assessment, Development, and Evaluation) approach was used to assess the quality of evidence, graded as high, moderate, low, or very low [29]. The initial GRADE quality score defaults to high and would then be downgraded according to pre-specified fields, including the risk of bias (over 20%), inconsistency (I^2^ > 50% and Pheterogeneity < 0.1), indirectness (presence of limitations to the universalization of the results), imprecision (95% confidence intervals (CIs) overlapped with the minimally important difference, i.e., 2 mmHg for blood pressure [30], 0.1 mol/L for blood lipids [31], 0.5% for A1C [32], 0.56 mmol/L or 10 mg/dL for FBG [33], and 5 pmol/L for FBI [34]), and publication bias (significant evidence of small study effects).

### 2.6. Statistical Analysis

The mean differences in specific cardiometabolic risk factors between intervention and control groups and the SDs were used as the basis for each trial comparison. The effect size was evaluated according to the Cochrane guidelines [27]. The random effects model was used to generate an effect size for cardiometabolic risk factors, expressed as the weighted mean difference and 95% CI. Heterogeneity was estimated among the included studies using I2 statistics. The significance for heterogeneity was set at *p* < 0.05, with an I2 > 50%, which was considered to be evidence of substantial heterogeneity [35,36]. Subgroup analyses were performed based on different cardiometabolic health statuses of the participants included in the trials. Possible publication bias was evaluated by visual inspection of funnel plots and Egger’s linear regression test (*p* < 0.05 indicates the presence of publication bias [37]). Sensitivity analyses were performed to assess the effect of individual studies on pooled effect sizes by omitting one study at a time [38]. RevMan (version 5.4) and Stata/SE (version 17.0) software were applied to all statistical analyses.

## 3. Results

### 3.1. Study Selection and Characteristics of Included Trials

The selection process of this study is summarized as a flow chart in Figure 1. A total of 5853 articles were identified through the combined search. Of the remaining 2545 articles, 2107 were identified as unrelated after reviewing for titles and abstracts, while 438 were reviewed and evaluated for suitability. Studies lacking randomization (*n* = 97), relevant outcomes (*n* = 136), or consisting of participants with severe disorders (*n* = 51) were excluded, leaving 130 articles (161 studies) from which to extract and analyze data. Characteristics of covered trials are outlined in Appendix A.

The current analysis involved 161 studies with a total of 12,307 participants aged 18–75 years (median age: 49.0 years). The included studies were conducted in participants intervened with n-3 fatty acid (*n* = 89), n-6 fatty acid (*n* = 23), n-9 fatty acid (*n* = 16), lycopene (*n* = 15), astaxanthin (*n* = 9), and beta-carotene (*n* = 6). The intervention lasted from 2 weeks to 5 years.

### 3.2. Effect of Antioxidant Lipid Supplementations on Blood Pressure

Ninety-seven eligible RCTs involving 8576 participants were included to investigate the effects of antioxidant lipid supplementations on blood pressure (Figure 2). Lycopene supplementation decreased systolic blood pressure (−1.95 [−3.54, −0.36] mmHg) in the total population. No other lipid supplementations displayed an effect on blood pressure in the total population.

### 3.3. Effect of Antioxidant Lipid Supplementations on Blood Lipids

A total of 119 eligible RCTs involving 8001 participants were included to assess the effects of antioxidant lipid supplementations on blood lipids (Figure 3). In the total population, n3 fatty acid significantly decreased TG (−0.37 [−0.47, −0.28] mmol/L); n6 fatty acid significantly improved LDL-C (−0.39 [−0.71, −0.06] mmol/L), HDL-C (0.08 [0.01, 0.14] mmol/L), and TC (−0.24 [−0.37, −0.11] mmol/L); astaxanthin significantly improved LDL-C (−0.11 [−0.21, −0.01] mmol/L), HDL-C (0.13 [0.05, 0.21] mmol/L), TC (−0.22 [−0.32, −0.12] mmol/L), and TG (−0.46 [−0.83, −0.10] mmol/L); beta-carotene significantly decreased TC (−0.13 [−0.23, −0.04] mmol/L); and notably, n9 fatty acid improved TG (0.06 [0.01, 0.11] mmol/L) in the total population.

### 3.4. Effect of Antioxidant Lipid Supplementations on Glycemic Status

Seventy-six eligible RCTs involving 4518 participants were included to assess the effect of antioxidant functional lipid supplementations on glycemic status (Figure 4). Astaxanthin improved FBI (−2.66 [−3.98, −1.34] pmol/L) in the total population. Notably, n9 fatty acid increased A1C (0.04 [0.02, 0.06] %) in the total population. No other lipid supplementations displayed an effect on glycemic status in the total population. 

### 3.5. Effect of Antioxidant Lipid Supplementations among Participants with Different Cardiometabolic Health Statuses

Subgroup analysis of antioxidant lipid supplementations was performed in participants with different cardiometabolic health statuses (Figure 5). For healthy populations, antioxidant functional lipid supplementations improved HDL-C (0.03 [0.00, 0.06] mmol/L) and TG (−0.15 [−0.25, −0.06] mmol/L). For participants with pre-diabetes or T2D, antioxidant lipid supplementations improved HDL-C (0.28 [0.07,0.50] mmol/L) and TG (−0.26 [−0.27, −0.15] mmol/L). For hypertensive participants, antioxidant lipid supplementations improved TC (−0.40 [−0.68, −0.12] mmol/L). For dyslipidemia participants, antioxidant lipid supplementations improved DBP (−0.81 [−1.51, −0.10] mmHg), HDL-C (0.16 [0.08, 0.24] mmol/L), and TG (−0.56 [−0.81, −0.30] mmol/L). For participants with overweight or obesity, antioxidant lipid supplementations improved HDL-C (0.06 [0.02, 0.10] mmol/L) and TG (−0.13 [−0.24, −0.02] mmol/L). Lastly, antioxidant lipid supplementations improved DBP (−3.00 [−4.15, −1.85] mmHg) in participants with metabolic syndrome.

## 4. Discussion

This study intended to provide a comprehensive analysis and identify appropriate antioxidant lipid supplementations to improve cardiometabolic health. Overall, our results suggest that n3 fatty acid, n6 fatty acid, lycopene, astaxanthin, and beta-carotene significantly improve cardiovascular risk factors, including SBP, all lipid profiles, and FBI in the total population (Figure 6). The benefits of antioxidant lipid supplementations appeared to be more pronounced in blood pressure and blood lipids in participants with different cardiometabolic health statuses. Particularly, n9 fatty acid was identified to increase TG and A1C in the total population in our results, which increases CVD risk and is not suggested to be over-consumed. This is probably attributed to excessive n9 fatty acid enhancing oxidative damage and inducing oxidative stress [39,40], though its specific molecular mechanism has not yet been elucidated.

It has been reported that dietary intervention focused on n6 and n3 fatty acids may improve cardiovascular risk factors [41], which is consistent with our results. The current meta-analysis still supports that n3 fatty acid supplementation improves HDL-C and TG levels, probably through regulating the coding of different genes involved in lipid metabolic hemostasis at transcriptional and post-transcriptional levels [42]. For example, consumption of n3 fatty acid suppresses the transcription of the sterol regulatory element-binding protein gene, thereby inhibiting the de novo synthesis of TGs [43]. Similarly, n6 fatty acid supplementation improves HDL-C and TC levels, probably by contributing to cholesterol catabolism through upregulating the expression of the cholesterol 7α-hydroxylase gene, which encodes an enzyme that regulates the pathway for cholesterol to bile acids, via peroxisome proliferator-activated receptors [44]. In this study, n9 supplementation was found to increase TGs and A1C, which is inconsistent with previous findings [45]. 

Lycopene supplementation decreased SBP, which is consistent with previous findings [46,47]. The antihypertensive properties of lycopene are thought to derive from the stimulation of nitric oxide production by the vascular endothelium [48]. Nitric oxide exerts a complex effect on the modulation of local and systemic vascular resistance, blood flow distribution, as well as arterial pressure [49]. Previous studies reported a beneficial effect of astaxanthin on preventing diabetes and atherosclerosis [22,50]. This analysis underscores the protective role of astaxanthin supplementation in improving lipid profiles and insulin metabolism. As for blood lipids, astaxanthin increases hepatic LDL receptor levels and sterol regulatory element-binding protein 2, which modulates cholesterol metabolism [51]. Moreover, astaxanthin could improve insulin secretion by modulating impaired glucose metabolism and beta cell dysfunction through glucose transporter 4 regulation [52,53]. 

In this study, beta-carotene supplementation showed a beneficial effect on TC in the total population. However, some studies demonstrated that beta-carotene had potentially harmful effects on CVD mortality and cancer incidence in smokers. The potential explanation is that beta-carotene offers several therapeutic effects as a provitamin A, including an antioxidant effect to neutralize reactive oxygen species and regulate connexin expression, thereby improving communication through gap junctions [54]. However, in some conditions, such as high oxygen concentrations in smokers, beta-carotene turns into a pro-oxidant, generating beta-carotene radical cations that require vitamin C for repair [55]. Due to low serum levels of vitamin C in smokers, beta-carotene radicals may contribute to an increased risk of CVDs and cancer [56]. 

The association between antioxidant supplementation and cardiovascular disease prevention has been controversial. To date, analyses of certain well-designed studies and some preclinical studies have led to the hypothesis that supplementation with pure free radical scavengers may not be sufficient to produce beneficial effects in pre-protective models [57]. Therefore, only compounds that are capable of influencing oxidative stress through more than one pathway or have pleiotropic properties can produce significant clinical effects. This is supported by a number of basic studies that suggest that the heavy use of certain free radical scavengers may exacerbate, rather than alleviate, oxidative stress [45,57]. The results of previous studies suggest that the preventive effect of supplements on cardiovascular disease will be proportional to the pleiotropic level of the substance administered [45]. Specifically, n-3 fatty acids, followed by folic acid and coenzyme Q10, showed beneficial effects, whereas scavengers with lower pleiotropic effects, such as vitamins C and E, did not alter the results, whereas some pure free radical scavengers with minimal cardiovascular pleiotropic effects, such as beta-carotene, worsened the results, which is also consistent with our findings.

## 5. Study Limitations and Conclusions

Several limitations need to be considered when interpreting results from current analyses. First, in this study, there was a lack of evidence regarding the improvement in cardiovascular outcome events and incidence of T2D, which leaves uncertainty as to whether antioxidant functional lipid supplementations also ameliorated the occurrence of CVD events. Second, the heterogeneity of included studies was high due to limitations of the number and sample size, variation in intervention duration between studies, wide study time range, as well as the low-quality evidence of some RCTs, although trials with intervention durations of less than 1 week were excluded from our analysis. Thirdly, owing to the lack of RCTs, some antioxidant functional lipid supplementations were not included in the current analysis, including dietary β-ketones with biologically relevant antioxidant activity and palmitoleic acid (16:1 n-7), known to reduce atherosclerotic lesions. Moreover, since the intervention period of the existing RCTs is short (median 12 weeks), it is unknown whether long-term administration will cause health hazards. At present, there are still many uncertainties and controversies regarding the recommendation of antioxidant functional lipid supplementations for the adjunct treatment of patients with T2D due to inconsistent and insufficient clinical results. Well-designed, high-quality, large, and long-term studies are needed to strengthen the existing evidence on the efficacy of antioxidant functional lipid supplementations in modulating cardiometabolic risk factors.

Ultimately, antioxidant functional lipid supplementations, except n9 fatty acid, exert beneficial effects on blood pressure, blood lipids, and glycemic parameters in the total population. The benefits of antioxidant functional lipid supplementations appeared to be most evidenced in blood pressure and blood lipids in participants with different cardiometabolic health statuses. Our findings highlight the importance of antioxidant functional lipid diversity and the balance of benefits and risks. Considerations should also be given to administering higher doses and longer durations when designing personalized intervention strategies aimed at enhancing cardiometabolic health. From the perspective of research, it should be noted that although current information opens up the prospect of consolidating the role of antioxidant lipids in preventive cardiology in the future, there is still a long way to go in providing evidence. In terms of routine clinical practice, these results are beginning to open up space for incorporating new tools into the therapeutic armory aimed at preventing cardiovascular disease in specific populations.

## Figures and Tables

**Figure 1 nutrients-16-02213-f001:**
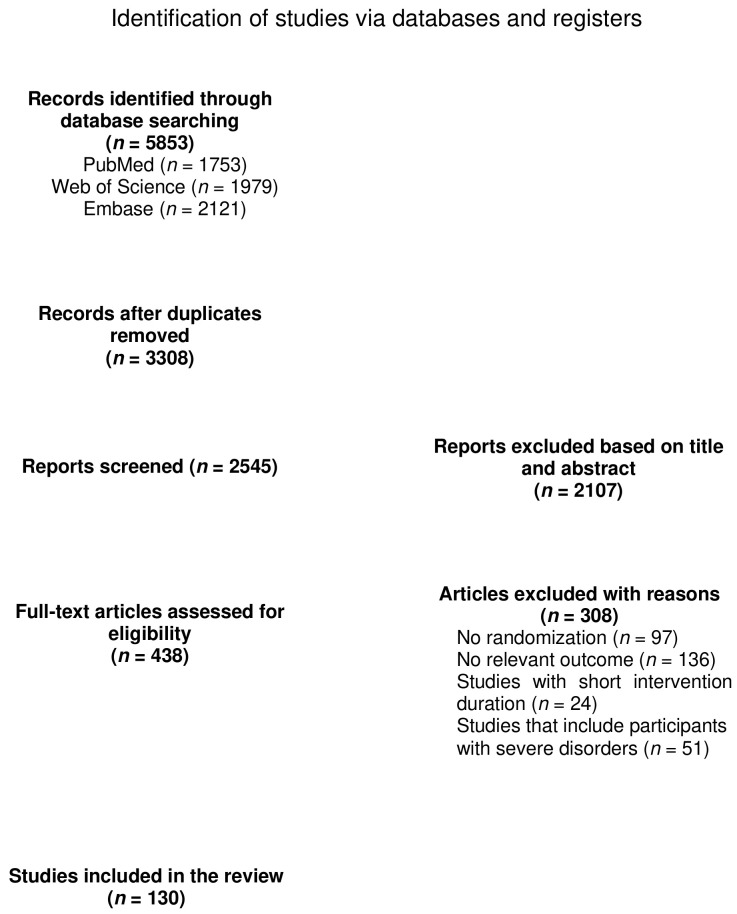
Flow chart of study selection.

**Figure 2 nutrients-16-02213-f002:**
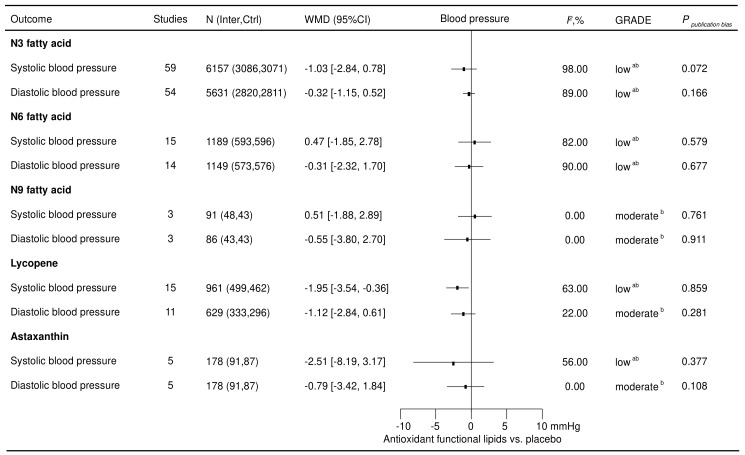
Effects of antioxidant lipid supplementations on blood pressure. Ctrl—control group; Inter—intervention group; SBP—systolic blood pressure; DBP—diastolic blood pressure; WMD—weighted mean difference; *I*^2^—values for between-study heterogeneity; ^a^—rated down for inconsistency; and ^b^—rated down for imprecision.

**Figure 3 nutrients-16-02213-f003:**
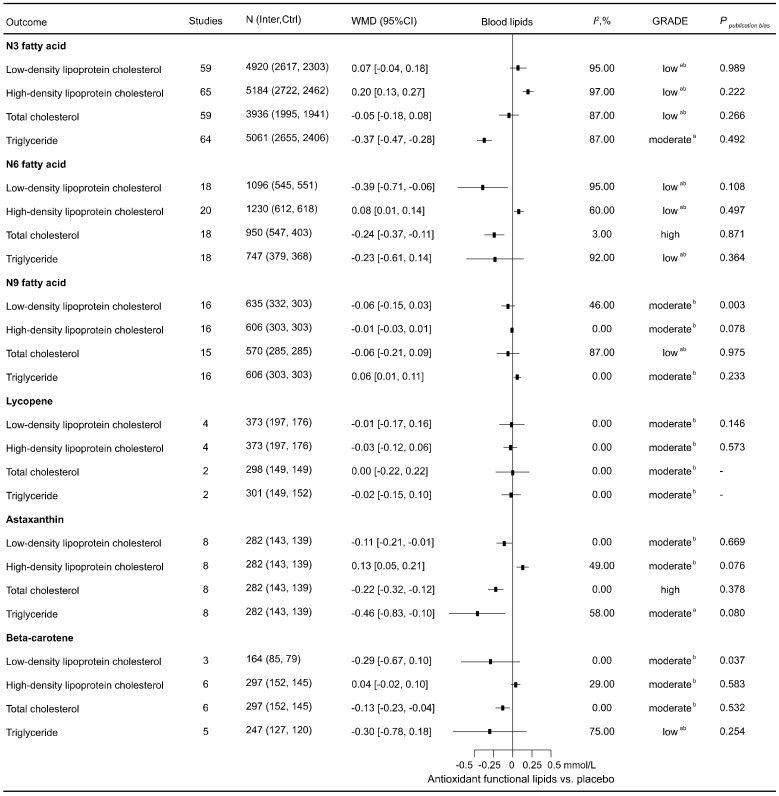
Effects of antioxidant lipid supplementations on blood pressure. Ctrl—control group; Inter—intervention group; SBP—systolic blood pressure; DBP—diastolic blood pressure; WMD—weighted mean difference; *I*^2^—values for between-study heterogeneity; ^a^—rated down for inconsistency; and ^b^—rated down for imprecision.

**Figure 4 nutrients-16-02213-f004:**
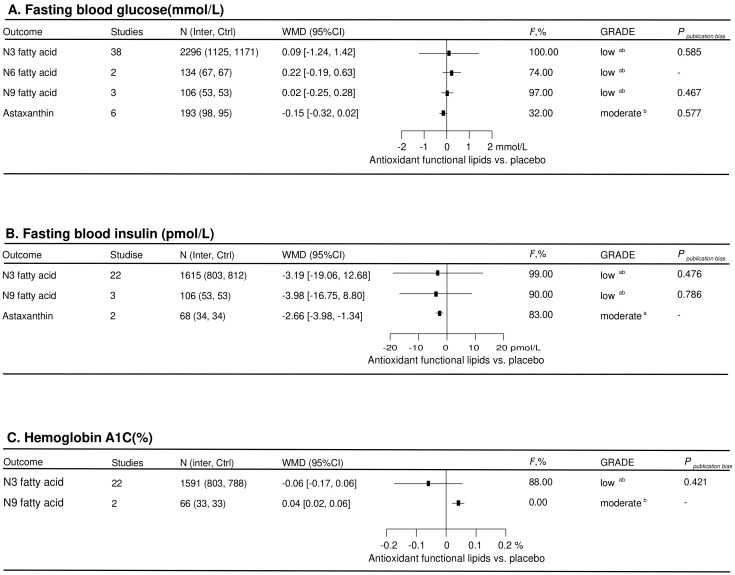
Effects of antioxidant lipid supplementations on glycemic control. Ctrl—control group; Inter—intervention group; FBG—fasting blood glucose; A1C—hemoglobin A1c; FBI—fasting blood insulin; WMD—weighted mean difference; *I*^2^—values for between-study heterogeneity; ^a^—rated down for inconsistency; and ^b^—rated down for imprecision.

**Figure 5 nutrients-16-02213-f005:**
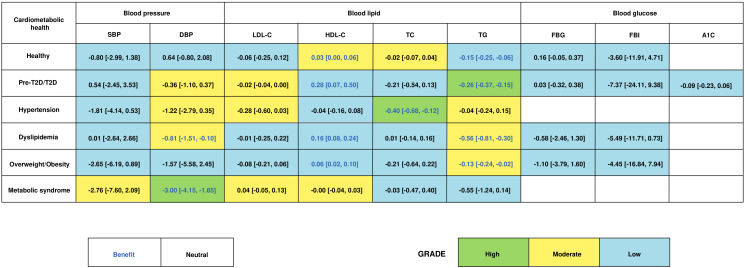
Effects of antioxidant lipid supplementations on participants with different cardiometabolic health statuses. Subgroup analysis of antioxidant lipid supplementations was performed in healthy participants and participants with pre-type 2 diabetes (Pre-T2D)/T2D, hypertension, dyslipidemia, overweight/obesity, or metabolic syndrome.

**Figure 6 nutrients-16-02213-f006:**
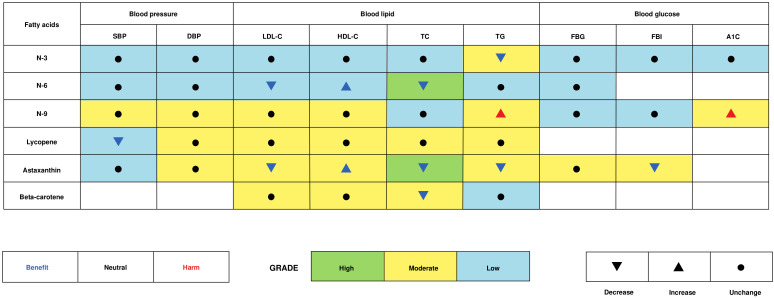
An evidence-based map of antioxidant lipid supplementations on cardiometabolic health.

## Data Availability

The original contributions presented in the study are included in the article/Appendix A, further inquiries can be directed to the corresponding authors.

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
