# Peer review of "Antioxidant Lipid Supplement on Cardiovascular Risk Factors: A Systematic Review and Meta-Analysis"

_nutrients, 2024, doi:10.3390/nu16142213_

Round 1

Reviewer 1 Report

Comments and Suggestions for Authors

Thank you very much for your interesting research. Some points must be carefully revised:

TITLE. Perhaps the word ‘functional’ should be deleted (‘antioxidant’ is enough to understand that they are functional). This suggestion might be applied to the whole text, for instance, in subsection 3.2, 3.3., 3.4., etc.

INTRODUCTION. 40-41. This statement must be replaced. ‘Blood glucose’ and ‘blood pressure’ are not risk factors per se (please, include ‘high levels of’ -or similar-). ‘Lipids’ are not a risk factor (high levels of specific lipids are a risk factor).

INTRODUCTION. More information about oxidative status, oxidative stress and dietary antioxidants must be included in this section.

RESULTS. Perhaps the subsection 3.1. might be part of the Materials & Methods section.

RESULTS. Figure 5: It should be revised since the results cannot be visualized because of the quality/resolution of the Figure.

DISCUSSION. Figure 6: Same issue than for Figure 5.

DISCUSSION. Which position do functional lipids place considering other dietary antioxidants? For instance, phenolic compounds. Comparisons with other molecules would reinforce your discussion.

CONCLUSION. A longer and more critical conclusion is recommended, emphasizing on current limitations of the state of the art and suggesting future perspectives. Perhaps section 5 and 6 can be merged.

Author Response

COMMENTS 1:TITLE. Perhaps the word ‘functional’ should be deleted (‘antioxidant’ is enough to understand that they are functional). This suggestion might be applied to the whole text, for instance, in subsection 3.2, 3.3., 3.4., etc.

RESPONSE: Thanks for the suggestion. The relevant section has been amended from ‘antioxidant functional lipid’ to ‘antioxidant lipid’.

COMMENTS 2:INTRODUCTION. 40-41. This statement must be replaced. ‘Blood glucose’ and ‘blood pressure’ are not risk factors per se (please, include ‘high levels of’ -or similar-). ‘Lipids’ are not a risk factor (high levels of specific lipids are a risk factor).

RESPONSE: Thanks for the suggestion. Inaccurate descriptions have been modified:

“High levels of blood glucose, blood pressure, and lipids, including low-density lipo-protein cholesterol, total cholesterol, and triglyceride, have been identified over the past 50 years as modifiable risk factors that act as predictors of CVD events and are thus key targets for the primary prevention of CVDs”

COMMENTS 3:INTRODUCTION. More information about oxidative status, oxidative stress and dietary antioxidants must be included in this section.

RESPONSE: Thanks for the suggestion. The descriptions of cardiovascular disease, oxidative stress, and dietary antioxidants have been added to the article:

“Oxidative stress is a contributing factor to CVD which has long been recognized as being involved in the pathogenesis of CVD (10). The adverse effects of oxidative stress on the cardiovascular system are attributed to decreased nitric oxide availability, inflammatory responses, and lipid peroxidation through which reactive oxygen species disrupt the homeostasis of the vascular wall, leading to defective endothelium-dependent vasodilatation (11). Once formed, ROS activates nuclear factor-kappa B, leading to transcriptional activation of more than 100 genes involved in the immune system and inflammatory response, such as tumor necrosis factor-α, interleukin-1β and interleukin-6. Macrophages absorb oxidized low-density lipoprotein and then convert to foam cells in vascular endothelial cells, leading to the development of atherosclerotic lesions (10, 11). Preclinical and epidemiological studies have suggested that dietary antioxidants such as vitamin E, carotenoids, and polyphenols inhibit oxidative stress, thereby providing cardiovascular protection.”

Reference:

  1. Hu Q.; Fang Z.; Ge J.; Li H. Nanotechnology for cardiovascular diseases. Innovation (Camb). 2022; 3(2):100214.
  2. Beijian Zhang WL.; Yun Cai.;Liwei Liu.;  Xiurui Ma.;  Wenlong Yang.;  Shu Meng.;  Gang Zhao.;  Aijun Sun.;  Junbo Ge. Global burden of adolescent and young adult cardiovascular diseases and risk factors: Results from Global Bur-den of Disease Study 2019. The Innovation Medicine. 2024; 2.

COMMENTS 4:RESULTS. Perhaps the subsection 3.1. might be part of the Materials & Methods section.

RESPONSE: Thanks for the suggestion. Subsection 3.1 presents primarily the results of the literature screening, rather than the screening methodology. Besides, since most meta-analyses are structured to present this section in the Results section, we preferred to display the section as it was originally presented.

COMMENTS 5:RESULTS. Figure 5: It should be revised since the results cannot be visualized because of the quality/resolution of the Figure.

RESPONSE: We apologize for the image clarity issue. The original image has been replaced with a higher-resolution image.

COMMENTS 6:DISCUSSION. Figure 6: Same issue than for Figure 5.

RESPONSE: We apologize for the image clarity issue. The original image has been replaced with a higher-resolution image.

COMMENTS 7:DISCUSSION. Which position do functional lipids place considering other dietary antioxidants? For instance, phenolic compounds. Comparisons with other molecules would reinforce your discussion.

RESPONSE: Thanks for the suggestion. A comparative discussion of relevant antioxidants has been added to the Discussion section:

“The association between antioxidant supplementation and cardiovascular disease prevention has been controversial. To date, analyses of certain well-designed studies and some preclinical studies have led to the hypothesis that supplementation with pure free radical scavengers may not be sufficient to produce beneficial effects in pre-protective models (57). Therefore, only compounds that are capable of influencing oxidative stress through more than one pathway or have pleiotropic properties can produce significant clinical effects. This is supported by a number of basic studies that suggest that the heavy use of certain free radical scavengers may exacerbate, rather than alleviate, oxidative stress (45, 57). The results of previous studies suggest that the preventive effect of supplements on cardiovascular disease will be proportional to the pleiotropic level of the substance administered (45). Specifically, n-3 fatty acids, followed by folic acid and co-enzyme Q10, showed beneficial effects, whereas scavengers with lower pleiotropic effects, such as vitamins C and E, did not alter the results, whereas some pure free radical scavengers with minimal cardiovascular pleiotropic effects, such as beta-carotene, worsened the results, which is also consistent with our findings.”

Reference:

  1. An P.; Wan S.; Luo Y.; et al. Micronutrient Supplementation to Reduce Cardiovascular Risk. J Am Coll Cardiol. 2022; 80(24):2269-85.
  2. Gormaz JG.; Carrasco R. Antioxidant Supplementation in Cardiovascular Prevention: New Challenges in the Face of New Evidence. J Am Coll Cardiol. 2022; 80(24):2286-8.

COMMENTS 8:CONCLUSION. A longer and more critical conclusion is recommended, emphasizing on current limitations of the state of the art and suggesting future perspectives. Perhaps section 5 and 6 can be merged.

RESPONSE: Thanks for the suggestion. Sections 5 and 6 have been merged. Besides, a more critical conclusion has been stated.

Reviewer 2 Report

Comments and Suggestions for Authors

The abstract is too long. It should have no more than 200 words.

The Introduction has to be expanded. More data about the antioxidant functional lipid supplements should be provided. In this section, the readers have to understand the need and justification for the present study. In its current state is too brief and superficial.

The Prisma flowchart well referenced needs to be provided.

More data on the included studies should be given in the Results section. I miss tables and further discussion in this systematic review manuscript.

The quality of figures (special figure 5) is poor.

The authors mention Appendix 1 on line 144 but it is not available.

References should be formatted according to the journal’s guidelines.

Author Response

COMMENTS 1:The abstract is too long. It should have no more than 200 words.

RESPONSE: Thanks for the suggestion. Abstract section has been shortened.

COMMENTS 2:The Introduction has to be expanded. More data about the antioxidant functional lipid supplements should be provided. In this section, the readers have to understand the need and justification for the present study. In its current state is too brief and superficial.

RESPONSE: Thanks for the suggestion. The article has been enriched with descriptions of functional lipid supplements:

“Functional lipids mainly include polyunsaturated fatty acids such as linoleic acid, lino-lenic acid, eicosapentaenoic acid, and eicosahexaenoic acid, as well as phytosterols (9). Polyunsaturated fatty acids are the mainstay of functional lipids research and are gen-erally defined as fatty acids containing either two or more double carbons with a carbon length of 18 or more. Phytosterols are active components that are widely found in roots, stems, leaves, fruits, and seeds of plants and possess favorable antioxidant properties and anti-inflammatory effects (9).”

Reference:

  1. Badriah Alabdulkarim ZANB.; Shaista Arzoo. Role of some functional lipids in preventing diseases and promoting health. Journal of King Saud University - Science. 2012; Volume 24(Volume 24):319-29.

COMMENTS 3:The Prisma flowchart well referenced needs to be provided.

RESPONSE: Thanks for the suggestion. The original Figure 1 has been replaced by the Prisma flowchart.

COMMENTS 4:More data on the included studies should be given in the Results section. I miss tables and further discussion in this systematic review manuscript.

RESPONSE: Thanks for the suggestion. Detailed information on the included studies is provided in Appendix 1 of the Supplementary Material, including the last name of the first author, publication year, geographic location of the study population, study design, participant characteristics (number of participants, mean age, sex, and health status), intervention substance, control substance, intervention dose, and intervention duration. Further discussion has been added to the article.

COMMENTS 5:The quality of figures (special figure 5) is poor.

RESPONSE: We apologize for the image clarity issue. The original image has been replaced with a higher-resolution image.

COMMENTS 6:The authors mention Appendix 1 on line 144 but it is not available.

RESPONSE: Appendix 1 can be found in Supplementary material, page 1.

COMMENTS 7:References should be formatted according to the journal’s guidelines.

RESPONSE: We apologize for the incorrect reference format. References have been formatted according to the journal’s guidelines.

Round 2

Reviewer 2 Report

Comments and Suggestions for Authors

A reference is missing in Figure 1.